# Cataract Surgery in Low-Income Countries: A Good Deal!

**DOI:** 10.3390/healthcare10122580

**Published:** 2022-12-19

**Authors:** Steffen Flessa

**Affiliations:** Department of Health Care Management, Faculty of Law and Economics, University of Greifswald, 17487 Greifswald, Germany; steffen.flessa@uni-greifswald.de; Tel.: +49-3834-420-24-77

**Keywords:** Africa, blindness, cataract, cataract surgery, health economics, low-income country, visual impairment

## Abstract

Cataract is a major cause of blindness worldwide. In particular, in low-income countries, the burden of disease as well as its direct and indirect economic cost are a major challenge for the population and economy. In many cases, it would be possible to prevent or cure blindness with a comparably simple cataract surgery, but many countries lack the resources to strengthen healthcare systems and implement broad cataract surgery programs reaching, in particular, the rural poor. In this paper, we analyse whether such an intervention could be cost-effective or even cost-saving for the respective health systems. We calculate the net value of the lifelong costs of cataract with and without surgery. This calculation includes direct costs (e.g., treatment, glasses, surgery) as well as indirect cost of the caregiver and the patient. We total all costs from the year of onset of cataract until death and discount the respective values to the year of onset. We define the surgery as cost-saving if the net-value of costs with surgery is lower than without surgery. If the cost per quality adjusted life year is lower than one gross national product per capita, we define the intervention as highly cost-effective. We find that the cost-effectiveness of cataract surgery depends on the age of onset of the disease and the age of surgery. If the surgery is performed with the beginning of severe impairment, even surgery of a 78-year-old patient is still cost-saving. Almost all possible constellations are highly cost-effective, only for the very old it is questionable whether the surgery should be performed. The simulations show that cataract surgery is one of the most cost-effective interventions. However, millions of people in low-income countries still have no chance to prevent or cure blindness due to limited resources. The findings of this paper clearly call for a stronger effort to reach poor and rural populations with this cost-effective service.

## 1. Introduction

Blindness and vision impairment are a major medical and economic problem worldwide. The ‘Lancet Global Health Commission on Global Eye Health: vision beyond 2020′ states that some 1.1 billion people are suffering from some kind of vision impairment [1]; other organisations estimate even higher numbers [2]. Roughly half of people living with vision impairment or blindness have distance vision impairment, the other half have near vision impairment [1] with presbyopia, cataract, refractive errors and glaucoma as the main causes. The prevalence of vision impairment is much higher in low- and middle-income than in high-income countries, and, in particular, prevention and cure of eye diseases is a major issue of concern in many least developed countries [3].

Cataract is a cloudy area in the lens leading to decreased vision [4]. It is estimated that about 50% of all blindness and 33% of visual impairment are caused by cataracts [1] although a comparably simple lens replacement (cataract surgery) can highly effectively restore vision in most cases. Consequently, cataract is mainly a medical problem of low- and middle-income countries to be faced by the vulnerable population, in particular, people living in rural areas, older people, women and the illiterate [5]. With the demographic transition of these countries, it is expected that the prevalence of cataracts will increase dramatically unless cataract surgery is made available even in rural and impoverished settings in Africa and Asia [6].

The tangible cost of blindness and vision impairment are consequently high and cover cost of treatment (direct provider and household cost) as well as opportunity cost. The latter occur because a person with reduced vision or blindness will not be as productive as a person without impairment. At the same time, a severely impaired or blind person will frequently need a caregiver to support him, who himself will face some opportunity cost of reduced productivity. As the ‘Lancet Global Health Commission on Global Eye Health: vision beyond 2020′ shows, the annual productivity loss due to visual impairment or blindness was highest in low- and middle-income countries (e.g., South Asia: 0.6% of GDP) and lowest in high-income countries (e.g., Western Europe: 0.15% of GDP) [1]. Marques et al. [7] calculated the global annual productivity loss due to moderate to severe visual impairment (MSVI) or blindness as 411 billion USD PPP or 0.3% of global GDP. The WHO estimates the costs of addressing the coverage gap of services to prevent or address MSVI and blindness. As Table 1 shows, the required resources to treat or prevent the respective conditions would be quite high, calling for a cost-effectiveness analysis in order to allocate scarce health care resources to those conditions with the highest efficiency.

A number of studies have shown the cost-effectiveness of cataract surgery, in particular in high-income countries [9,10]. However, our knowledge about the economic dimension of cataract treatment in low-income countries is limited [11,12]. Lansingh et al. calculated the cost-effectiveness for “developed countries” as 730 to 2400 international dollars per disability-adjusted life year (DALY) averted; the respective figures for “developing countries” ranged from 90 to 370 international dollars per DALY averted [13]. However, the composition of costs and the relevance of the age of the patient for the cost-effectiveness analysis has not been addressed appropriately.

This paper fills this research gap and presents a health economic analysis that allows assessing the costs, saved costs and gain of quality of life in order to calculate the cost-effectiveness of cataract surgery in a low-income country with respect to the age of the patient. The analysis is carried out from two separate perspectives, i.e., the financer and the private household. Thus, it includes direct (treatment, surgery, …) and indirect cost (e.g., caregiver). The time-horizon is the lifetime, i.e., we assess the time between onset of the disease and death of the patient.

## 2. Materials and Methods

We calculate the net-value of the total direct and indirect lifetime cost of a patient with cataract from the onset of the disease to death for the case with and without a cataract surgery at a certain age (from 40 to 98 years). If the net-value of the lifetime cost with surgery is lower than without the surgery, the intervention is called cost-saving. Otherwise, the difference is the net-value of the additional cost. The quotient between the net-value of this additional cost and the net-value of the additional quality of life is the cost-effectiveness, i.e., it shows how much has to be invested to gain one quality-of-life year. If this cost-effectiveness ration is lower than or equal to the average gross domestic product (GDP) per capita, we call an intervention highly cost-effective. If it is higher than the average GDP p.c. but lower or equal to twice the GDP p.c. we call it cost-effective [14].

### 2.1. Cost Functions

Modelling is always a balance between the desirable degree of precision and reality, on one hand, while keeping the model as simple and transparent as possible without pretending a degree of precision which does not exist. For instance, it would be ideal to make an agent-based model of a single patient with cataract and follow his lifetime process from onset to death [15]. This would require an enormous amount of data, in particular, of the individual disease progression, mortality and behaviour. In the absence of these data, each individual of the agent-based simulation would have the same characteristics, so that an “average patient” is modelled. Consequently, the model gives the impression that it is very precise while the data do not fulfil this claim. The same statement is true for a Markov model, which requires certain transition probabilities in order to simulate a cohort of patients. These probabilities are unknown for most low-income countries. For instance, the rest-of-life expectancy is documented, but not the age-specific mortality rates for each year of life.

Consequently, we decided to use a simple but very transparent model to assess the cost-effectiveness of the intervention. We calculate the net-value of lifetime costs with and without the intervention as the total of discounted direct and indirect costs per year between onset of the disease and the expected death. Consequently, we can only calculate the average cost without a distribution, but this should suffice to obtain an impression of the economic benefits of cataract surgery in low-income countries.

The cost function includes the costs of the surgical intervention, the annual cost of the treatment without operation, the productivity loss of the patient and the productivity loss of the caregiver. For this purpose, we define the following variables and constants.
  **Variables***NC_a_*Net value of cost of cataract of a person with onset age a, a = 40..99*CT_a_*Net value of cost of treatment of cataract of a person with onset age a, a = 40..99*CP_a_*Net value of productivity loss of a person with cataract with onset age a, a = 40..99*CC_a_*Net value of productivity loss of the caregiver for a person with cataract with onset age a, a = 40..99*NS_a_*Net value of cost of cataract surgery of a person with onset age a, a = 40..99*NQ_a_*Net value of quality of life of a person with cataract with onset age of a, a = 40..99

  **Constants**
*CTM*
Cost of treatment of cataract per year of mild/moderate impairment
*CTS*
Cost of treatment of cataract per year of severe impairment
*CTB*
Cost of treatment of cataract per year of blindness
*L_a_*
Life expectancy of a person in age a, a = 40..99
*t_m_*
Duration of cataract with mild/moderate impairment
*t_s_*
Duration of cataract with severe impairment
*t_b_*
Duration of cataract with blindness
*r*
Interest rate
*CPM*
Productivity loss per year for a person with cataract with mild/moderate impairment
*CPS*
Productivity loss per year for a person with cataract with severe impairment
*CPB*
Productivity loss per year for a person with cataract with blindness 
*CCM*
Productivity loss of a caregiver per year for a person with cataract with mild/moderate impairment
*CCS*
Productivity loss of a caregiver per year for a person with cataract with severe impairment
*CCB*
Productivity loss of a caregiver per year for a person with cataract with blindness
*ß_t_*
$\beta_{t}=\{\begin{array}{ll} 1 & if\;t<P\\ 0 & else \end{array}$ (dummy variable for pension age)
*P*
Pension age
*s*
Year of surgery
*CS*
Cost of surgery
*Qm*
Quality of life of a person with cataract with mild/moderate impairment
*Qs*
Quality of life of a person with cataract with severe impairment
*Qb*
Quality of life of a person with cataract with blindness

The model distinguishes the following stages based on the WHO definition [2]:-Mild or moderate impairment: visual acuity worse than 6/12 to 6/60.-Severe impairment: visual acuity worse than 6/60 to 3/60-Blindness: visual acuity worse than 3/60

The cost of treatment of cataract differs between the stages. We calculate the net value of the treatment cost as:$$CT_{a}=\overset{a+t_{m}-1}{\underset{t=a}{\sum }}CTM\cdot {\left(1+\frac{r}{100}\right)}^{a-t}+\overset{a+t_{m}+t_{s}-1}{\underset{t=a+t_{m}}{\sum }}CTS\cdot {\left(1+\frac{r}{100}\right)}^{a-t}+\overset{L_{a}-1}{\underset{t=a+t_{m}+t_{s}}{\sum }}CTB\cdot {\left(1+\frac{r}{100}\right)}^{a-t}$$

The productivity loss of a person with cataract depends again on the stage. The respective net value is calculated as:$$CP_{a}=\overset{a+t_{m}-1}{\underset{t=a}{\sum }}\beta_{t}\cdot CTM\cdot {\left(1+\frac{r}{100}\right)}^{a-t}+\overset{a+t_{m}+t_{s}-1}{\underset{t=a+t_{m}}{\sum }}\beta_{t}\cdot CTS\cdot {\left(1+\frac{r}{100}\right)}^{a-t}+\overset{L_{a}-1}{\underset{t=a+t_{m}+t_{s}}{\sum }}\beta_{t}\cdot CTB\cdot {\left(1+\frac{r}{100}\right)}^{a-t}$$

The productivity loss of the caregiver does also depend on the stage, and the net value is calculated as:$$CC_{a}=\overset{a+t_{m}-1}{\underset{t=a}{\sum }}CCM\cdot {\left(1+\frac{r}{100}\right)}^{a-t}+\overset{a+t_{m}+t_{s}-1}{\underset{t=a+t_{m}}{\sum }}CCS\cdot {\left(1+\frac{r}{100}\right)}^{a-t}+\overset{L_{a}-1}{\underset{t=a+t_{m}+t_{s}}{\sum }}CCB\cdot {\left(1+\frac{r}{100}\right)}^{a-t}$$

Consequently, the total cost of a patient with cataract without surgery is calculated as
$$NC=CT_{a}+CP_{a}+CC_{a}$$

If we assume that the surgery is performed in the year of life s, the net value of the cost of surgery is:$$NS_{a}=CS\cdot {\left(1+\frac{r}{100}\right)}^{a-s}$$

The total cost of a cataract patient with age a has to add the treatment cost as well as the productivity loss of the patient and the caregiver between the age of onset and the age of the surgery.

### 2.2. Quality of Life

We calculate the quality-adjusted life years (QALY) as a generic measure of the burden of cataract with and without surgery. For each year between the onset of the disease and death, the patient has a certain health status with a certain quality. The number of QALYs is calculated by discounting this quality of life and adding the respective values between onset and death. Consequently, we calculate the net value of the quality of life of a patient with cataract by adding the net values of the quality of life for the three different stages:$$NQ_{a}=\overset{a+t_{m}-1}{\underset{t=a}{\sum }}QM\cdot {\left(1+\frac{r}{100}\right)}^{a-t}+\overset{a+t_{m}+t_{s}-1}{\underset{t=a+t_{m}}{\sum }}QS\cdot {\left(1+\frac{r}{100}\right)}^{a-t}+\overset{L_{a}-1}{\underset{t=a+t_{m}+t_{s}}{\sum }}WB\cdot {\left(1+\frac{r}{100}\right)}^{a-t}$$

If a patient is operated on in the year of life s, this formula is applied for the year *a* to *s*−1, afterwards, the net value of the quality of life of restored vision is used.

### 2.3. Data

The constants listed in Section 2.1 are presented in Table 2. The quality of life statistics were taken from the global burden of diseases database [16], demographic and social statistics from other databases, e.g., the World Report on Vision, the World Development Indicators [17] and the World Data Atlas [18]. Parameters on productivity loss were retrieved from the literature [1,7], while the cost of surgery is based on literature [19,20,21] and own calculations. For estimating the treatment cost, we assumed a rural setting in Tanzania (where the author has practical experience).

Some parameters could not be retrieved from databases and literature, either because it does not exist at all or because the literature focusses on developed countries and there is a risk that these data are not representative for rural Africa. Consequently, we had to make estimates based on a mini-survey. For this purpose, we attended the annual meeting of the “Committee of the Prevention of Blindness” (April 2022, Wuerzburg, Germany) and gave out a questionnaire requesting estimates of the participants which could not be retrieved from other sources (see Appendix A). The members of this committee are experienced ophthalmologists with working experience in rural Africa so that they were appropriate experts to estimate the parameters. Seven participated and gave rather similar answers.

For instance, the onset of the disease in low-income countries is frequently earlier than in high-income countries reflecting the lower expectancy of life. The experts gave estimates ranging from 50 to 60 years with a mode of 55. They finally agreed that 55 would be a good estimate for this constant. Based on these different sources, Table 2 shows the respective parameters for the basic scenario.

The explanation of the process of retrieving the data of Table 2 also explains the characteristics of this research: We did not collect prime data in a particular site (e.g., hospital, program) but calculated the cost and cost-effectiveness based on estimates. The focus is not on a study design but on a methodology that allows to assess the economics of this intervention even under great uncertainty of data.

## 3. Results

The baseline scenario assumes the parameters shown in Table 2 with the supposition that the surgery is performed at the beginning of the second stage (severe visual impairment), 10 years after the onset of cataract with mild or moderate impairment. Based on the estimates of the experts, it also assumes that the patient is 60 years of age when he is operated on, i.e., all costs and benefits are discounted for a 50-year-old person. The 60-year-old patient has a life expectancy of 77 years. As shown in Table 3, if not operated on, the net-value of the total cost is 2048.45 USD; with the surgery, it is 801.47 USD, i.e., the cataract surgery is cost-saving. A total of 68% of costs is the opportunity cost of the caregiver, as the patient will be blind for 12 years. A total of 22% is the productivity loss of the patient, as he will have a reduced productivity until he retires (in the year of the surgery). The remaining 11% are the direct treatment costs. The main difference between the patient with and without the surgery is the productivity loss of the caregiver which is, in particular, high when the patient is blind, which will be avoided completely by the surgery when severe visual impairment starts.

Consequently, the surgery for the baseline scenario is cost-saving, but this depends on a number of variables. Figure 1 shows the tornado diagram for cost-savings. The respective constants are decreased and increased by 50%. Merely the pension age varies between 55–60–65 years. The chart shows that the parameters CPB, CPS, CPM and CTM have no impact on the cost-savings of the base case. The patient has the onset of the mild/moderate impairment at age 50 and develops severe impairment at age 60. The same year, he retires. Consequently, the costs of treatment and productivity loss in the phase of mild/moderate impairment (CTM, CPM) do not differ whether the patient is operated on or not in the age of 60. As he retires at the age of 60, he will not have any productivity loss beyond that age, irrespective of whether he is operated on or not. Certainly, the figures would alter if we make different scenarios, e.g., a person with onset age of 45 and a pension age of 65 with an operation at age 55 will have productivity gains due to the operation.

Figure 1 also shows that the highest impact on the cost-savings has the GDP p.c. In very poor countries, the cost-savings will be lower, in richer countries the cost-savings will be higher due to the opportunity cost of (the patient and) the caregiver. Another high impact has the change of the productivity loss of the caregiver, in particular, when the patient is blind (CCB). A reduction of the pension age will not make a difference, but an increase will also save more costs. The duration of the stages (mild/moderate, severe) does also determine the cost-savings. Longer durations will increase the cost-savings. If a patient stays, for instance, longer in the stage of mild/moderate impairment, he will—with a given life expectation—be fewer years blind. Thus, a longer duration in lower stages leads to lower lifetime costs while the cost of the surgery remains the same. Finally, the interest rate does also have high impact on the cost-savings. If the interest rates are higher, the net-value of the lifetime costs of blindness will be lower and thus the cost-savings will decrease.

Figure 2 shows the share of the components of cost-savings for three different scenarios:(1)Year of Surgery = Year of onset(2)Year of Surgery = First year of severe visual impairment(3)Year of Surgery = First year of blindness

For instance, the bar of “40-Onset” shows the cost components with onset of cataract in the age of 40 if he is operated on in the year of onset. It is obvious that the productivity loss of the patient plays only a role if the onset and the surgery are rather early. Otherwise, the patient will have retired when he is operated on so that there are no cost-savings. If the patient is operated on rather early there are cost-savings of treatment costs, otherwise the main cost is the productivity cost of the caregiver.

Furthermore, it is relevant to ask how the time of surgery determines the cost-savings. Figure 3 shows the result for the three scenarios. A later surgery in life can have two reasons. Firstly, because the onset of the cataract is later in life, and secondly because the surgery is postponed. The figure shows that the gain in quality of life is lower for both reasons, i.e., an early surgery results in higher increase in quality of life. At the same time, the cost-savings are lower if the patient is operated on early. For instance, if a patient has the onset of cataract at 80 years, the additional lifetime costs are 225.54 USD with an increase of quality of life of 0.909. The 80-year-old patient has a life expectancy of 5 years, and he will not live long enough to develop severe visual impairment due to cataract. He retires, so that no productivity loss is experienced, and with mild/moderate impairment, he will not require a caregiver. Thus, without an operation he will have only low costs. At the same time, an 80-year-old blind person will have high opportunity costs for the caregiver so that the surgery saves 19.24 USD and improves the quality of life by 0.19 QALYs. Consequently, there is good reason to perform the surgery for all blind patients irrespective of their age. For patients with mild/moderate impairment, it is cost-saving to perform the surgery before the age of 65; for patients with severe impairment it is cost-saving to operate on the patient until age 78.

For the case of surgery in the beginning of the severe impairment, the cost-saving corridor extents to a patient with onset of cataract at the age of 68 and a surgery at the age of 78. For older patients it is not cost-saving, e.g., the onset at the age of 69 leads to a net-value of additional lifetime costs of 21.95 USD and a gain of 0.69 QALYs. Still, the operation is highly cost-effective with cost of 32.01 USD per QALY. If the onset and the surgery are later, the cost-effectiveness will decrease, e.g., the costs per QALY are 117.21, 287.81 and 800.00 USD for onset age of 75, 80 and 85. For patients older than 95 years, the remaining rest-of-life expectancy is lower than one year; thus, it is insufficient to have any gain of quality of life.

If the surgery is performed in the same year as the onset of cataract, the corridor of cost-saving is smaller. A patient with onset age of 65 years (surgery age of 75) is not cost-saving any more. However, the cost per QALY is only 2.88 USD at this age. The statistic increases progressively. If the surgery is performed at the age of 90, the costs are 668.29 USD per QALY; if surgery is performed at the age of 92, it is 1400.00 USD per QALY. Finally, if the surgery is performed when blindness is fully developed, the procedure remains cost-saving until the onset age of 76 (surgery age of 91). At the onset age of 77, the cost per QALY is 100.00 USD. However, we have to state that the statistics for the three alternatives cannot be compared easily because the life expectancy is different. For instance, a person who is operated on at age 65 (i.e., at beginning of mild/moderate impairment) has a life expectancy of 78.8 years; a person who is operated on at age 75 (i.e., at the beginning of severe impairment) has a life expectancy of 82.5 years; a person who receives surgery at the age of 80 (i.e., at the beginning of blindness) has a life expectancy of 85.0 years. Thus, the longer we wait for the surgery, the higher is the likelihood that the patient will age and thus, the more cost-saving the surgery will be.

## 4. Discussion

This analysis shows that cataract surgery in low-income countries is cost-saving in most cases and still cost-effective for almost all other surgeries. However, this statement is only true if we assume that the caregiver is still of working age. As this is not always the case, the figures given above might over-estimate the cost saving. Furthermore, the biggest share of cost is opportunity cost for the household. These households have a good reason to invest in the operation themselves, but in poor and rural areas of low-income countries, they still might not be able to afford the surgery. Consequently, the service has to be sponsored by governments or charities, i.e., the costs are with these institutions while the benefits are with the households or the society. This might lead to imperfect decision-making.

This research adds to the literature by showing the cost-savings and cost-effectiveness, not only for one average patient with an average onset age and average age of surgery, but also for many age-sets. In comparison to other studies, our results show a higher cost-effectiveness, which is mainly due to the fact that the opportunity cost of caregivers was usually not included in other studies (e.g., [10,13]).

Based on this analysis we can state that cataract surgery in low-income countries is “a good deal”. Figure 4 shows the return-on-investment of the surgery if the year of surgery is the year of onset. Even for the “average” case of a patient with 60 years of age on onset and surgery, the return is still 12%. This statistic is calculated by changing the discounting factor of the respective formulae (r) until the cost-saving is zero. In other words, from an economic perspective it is definitely worthwhile investing in cataract surgery. However, this statement is only true if the health policy-maker cares about the indirect cost of the household. If we focus only on the perspective of the financer, the cost-effectiveness might be lower.

Finally, we have to ask for the conditions that a surgery is (1) not cost-saving or (2) not cost-effective anymore. A first answer is given by Figure 3, showing that the age of onset and the age of surgery are the most crucial conditions. The earlier a person develops cataract and is operated on, the more efficient the surgical intervention is. However, other parameters also influence cost-savings and cost-effectiveness. Table 4 shows the thresholds of parameters (for surgery in the year of onset), i.e., we change the parameters of the baseline model until the surgery is not cost-saving or (highly) cost-effective any more. The calculation is done for a person who is 60 years when mild/moderate impairment due to cataract starts and is operated on in the first year.

A first result is that the cost of treatment can be reduced to zero, but the surgery remains cost-saving. Instead of a net-value of cost-savings of 329.73 USD, the respective value is 111.76 USD. Originally, the life expectancy of a 60-year-old person is 77. If we reduce the life expectancy to 73, the surgery is not cost-saving any more. However, it remains highly cost-effective (i.e., with a cost-effectiveness ratio of less than 1 GDP p.c. per QALY) for a life expectancy of 61, i.e., even if the person will live only for one more year, it is cost-effective to perform the surgery.

The duration of the stages does also have an impact on the corridors. If the duration with mild/moderate impairment is increased to 15 years, the surgery is not cost-effective anymore, but the parameter cannot be increased so much that the intervention is not cost-effective. The duration of cataract with severe impairment will not have an impact on the corridors, i.e., even if we increase this parameter strongly, the cost-effectiveness corridor will not be left. If the GDP p.c. is reduced to 220 USD, the surgery will not be cost-saving anymore. Only for the completely unrealistic GDP p.c. of 25 USD the intervention will not be cost-effective.

A variation of the productivity loss of the patient will not have an impact on this case because the person will retire in the same year as the onset of the disease. The same is true for single variations of the productivity loss of the caregiver. Only if we reduce the productivity loss for severe impairment and blindness (e.g., 10 USD p.a. and 58 USD p.a.) the intervention is not cost-saving anymore. The pension age will not make a difference. Only if it were earlier, it would reduce the cost-savings, but the patient here is already 60 years old.

If the cost of the surgery increases to 630 USD, it will be not cost-saving. If the cost is more than 1799 USD it will not be highly cost-effective, but this amount is very unrealistic for a country with a GDP p.c. of 1100 USD.

Figure 5 summarizes the findings. For most ages of onset and surgery, as well as for all relevant parameter variations, cataract surgery is cost-saving in low-income countries. For patients with an onset of the mild/moderate impairment at age between 65 and 70, the intervention is not cost-saving, but still highly cost-effective. Only for very high ages, it is “only” cost-effective. The only case where cataract surgery in low-income countries is not cost-effective at a threshold of 2∙GDP p.c. is if the rest-of-life expectancy is less than one year with an age of 93 or higher. However, these cases are so rare in low-income countries that each individual has to be assessed separately. Generally, this analysis confirms that cataract surgery in low-income countries is a good deal.

It is also a matter of fact that financing cataract surgery in low-income countries is only one element of overcoming blindness. Even middle-income countries are still struggling to reach the entire population and overcome barriers, i.e., uptake and coverage of cataract surgery in low- and middle-income countries are a challenge as limited accessibility, poor quality of services, cultural habits and beliefs make it difficult to cover the population in need, even if financial resources are sufficient [22,23]. In a systematic review, Mailu et al. identified factors reducing the uptake of cataract surgery [24]. Economic parameters on the side of the demander as well as the supplier play a major role, e.g., socio-economic characteristics, costs of surgery, distance to the health facility and perceived quality of services.

Economic, social, demographic and cultural barriers lead to a situation where effective cataract surgical coverage (as the number of people in a population who have been operated on for cataract with a good outcome divided by the number of people operated on or requiring surgery) can be as low as 3.8% (Guinea Bissau) with a median of 14.8% in low-income countries [25]. Consequently, there is general agreement that there is a need to invest more into cataract surgery, and costs are not the only prohibiting factor, but an important one [26]. However, the actual cost of the respective programs and interventions are unknown or obsolete [10,27,28] for most low-income countries and respective estimates of cost-effectiveness based on these older publications are outdated themselves [29,30].

This paper gives estimates of actual direct and indirect costs and the resulting cost-effectiveness of cataract surgery in low-income countries. The respective findings are in line with other studies pointing at a high cost-effectiveness of cataract surgery. For instance, Lansingh et al. calculated the cost-effectiveness for “developing countries” as 90 to 370 international dollars per DALY averted [13] which is also demonstrating high cost-effectiveness of the intervention. However, their calculation was from the year 2007, and this clear evidence did not inspire a worldwide campaign to fight cataract in low-income countries. The evidence of our calculation and of other studies calls for a worldwide effort to end the avoidable blindness due to cataract—but this finding must be translated into a global policy.

The findings of this paper must be seen in the light of different types of limitations of this study.

Limitations of data. There are a number of limitations of this study due to missing or unreliable data while some data might be wrong for specific situations. Although the sensitivity analyses show that the parameters might change strongly without having a major impact on the thresholds of cost-savings and cost-effectiveness, there is still a possibility that the parameters of a real situation might be very different. For instance, our model does not apply to African patients flying for cataract surgery to Arabic countries as the costs of this surgery are much higher than what we assumed here.Furthermore, costs and outcomes depend on the professionalism of services provided. Poor quality of services will result in a completely different result of the health economic analysis of cataract surgery in low-income countries. However, the vast majority of cataract surgeries in these countries are performed professionally and have a tremendous result on the visual strength of the patient and his ability to live an independent and satisfying life—and they are cost-saving or cost-effective in almost all circumstances.Limitations of methodology: It would be ideal to use distributions and more advanced modelling techniques to produce more than averages. Stochastic models, such as Markov, discrete event simulation and agent-based simulations have the capacity to produce these distributions and allow a risk assessment. However, with the limited data given in the literature, such models could not be applied. Instead, they would pretend a degree of precision which does not exist. There is a great need to conduct more research in this field in order to produce data which might support more advanced modelling techniques in future.Another consequence of poor data is that a probabilistic sensitivity analysis would not really provide new insights. Usually, one would expect that a distribution of parameters can be taken from the literature, permitting an analysis of the impact of randomly varied parameters on major outcomes. However, in the absence of reliable data, this method cannot be applied as it would pretend a degree of precision which does not exist.Simplifying assumptions: We made some assumptions which could be challenged. We assume that the quality of life in the year of surgery is not reduced by the procedure. Thus, we assume that the surgery is successful and has no complications. As there might be severe problems associated with the surgery, we might over-estimate the gain of quality of life due to cataract surgery.

Consequently, the analysis presented in this paper is challenged by a high degree of uncertainty. We face a dilemma: on the one hand, it is necessary to calculate the cost and cost-effectiveness of cataract surgery in low-income countries in order to advise national and international intervention programs. On the other hand, many medical, economic, service and social parameters are uncertain. This is not unusual for health economic evaluations, but it is even stronger under conditions of low-income countries. Consequently, all results of this model must be handled with great caution. However, there is still a lot to learn from these calculations as they are “modelling for insights, not for numbers” [31]. The main result that cataract surgery in low-income countries is cost-effective or even cost-saving is a robust insight which seems not to depend on the uncertainty of structures and parameters.

## 5. Conclusions

This paper demonstrates that cataract surgery in low-income countries is cost-saving under most conditions and highly cost-effective for all relevant conditions. National cataract surgery programs will have a major impact on the quality of life of patients with a higher cost-effectiveness than many other interventions. Thus, we can conclude that national cataract surgery programs will pay off the investment, in particular, by freeing caregivers from giving attention to the blind and severely visually impaired. They should be financed sufficiently and administered professionally, in particular, to reach the population in rural and remote areas where cataract surgery is still unavailable in most hospitals.

However, we must be aware that financing the health care providers might not be sufficient. Grimes et al. analyzed the barriers to surgical care in low-income countries and conclude that the direct (e.g., cost of cataract surgery) are not the only factors prohibiting patients from seeking health care [23]. Other factors, such as distance, transport costs, poor roads, lack of suitable transport, and fear of anaesthesia are relevant factors as well. Thus, there is a need to embed cataract surgery in a comprehensive improvement of health care services, public transport and health education in order to reach the rural poor of all ages and educational levels.

## Figures and Tables

**Figure 1 healthcare-10-02580-f001:**
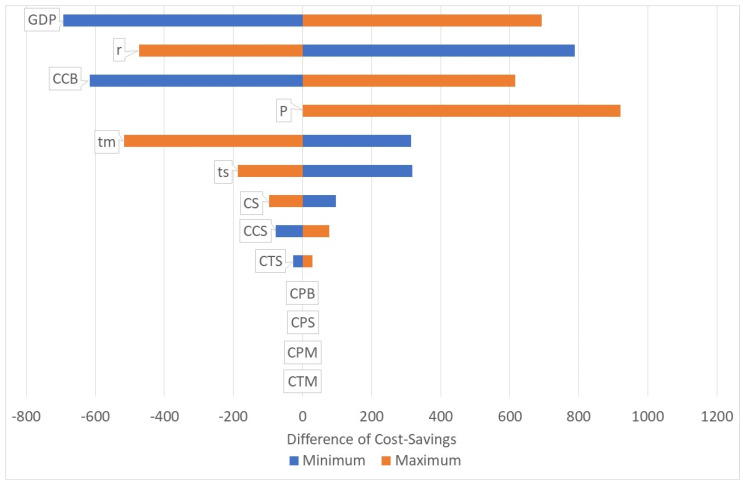
Tornado diagram. Source: own.

**Figure 2 healthcare-10-02580-f002:**
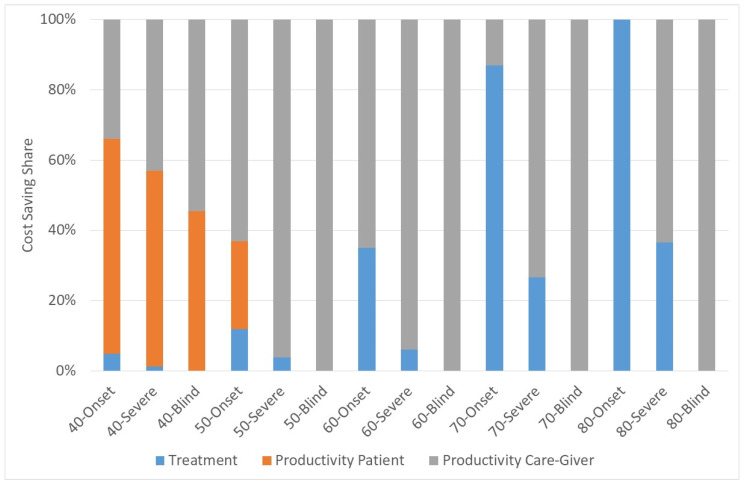
Share of cost-savings. Source: own.

**Figure 3 healthcare-10-02580-f003:**
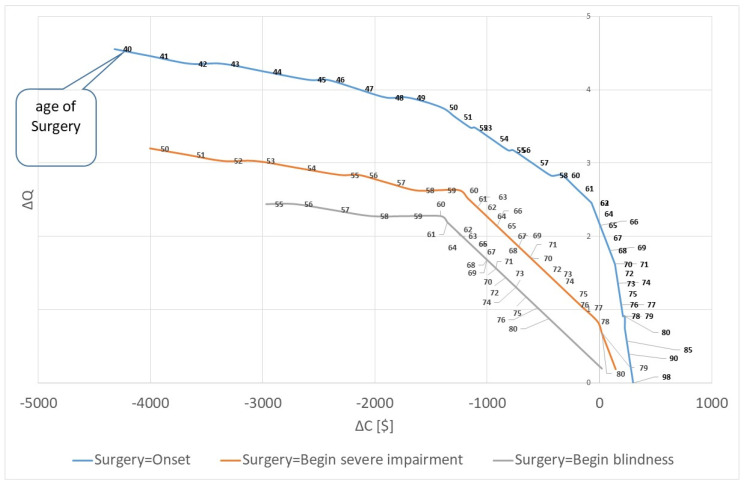
Cost-saving and quality of life: Marginal cost and marginal quality of life for different ages of surgery. Source: own.

**Figure 4 healthcare-10-02580-f004:**
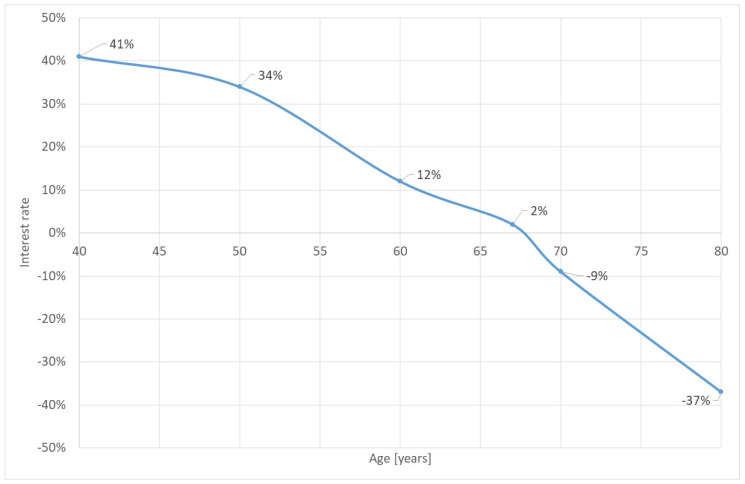
Return-on-investment of cataract surgery for surgery on onset. Source: own.

**Figure 5 healthcare-10-02580-f005:**
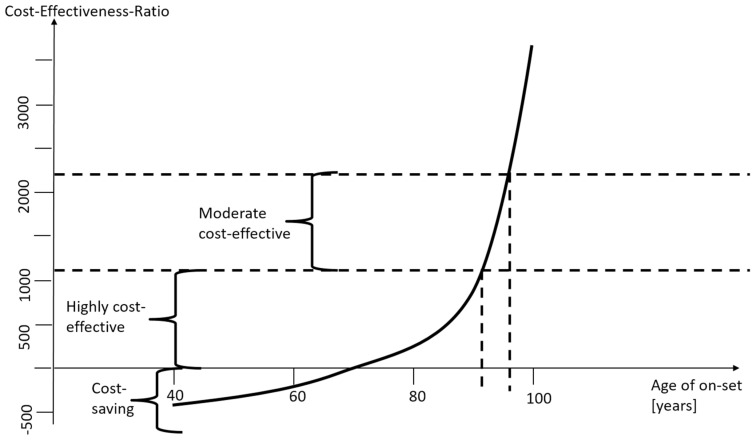
Cost-saving and cost-effectiveness corridor. Source: own.

**Table 1 healthcare-10-02580-t001:** Costs of addressing the coverage gap of visual impairment and blindness. Source: [8].

MSVI or Blindness Causes That Are…	Disease	USD (2018)
… treatable or addressable	Cataract	8,768,759,000
	Unaddressed Refractive Error (Distance)	6,988,223,000
	Unaddressed Refractive Error (Near)	9,035,476,000
	Total	24,792,458,000
… preventable	Diabetic Retinopathy	19,858,251,000
	Trachoma	494,077,000
	Glaucoma	11,744,642,000
	Total	32,096,970,000

**Table 2 healthcare-10-02580-t002:** Parameters.

Variable	Definition	Value	Source
*CTM*	Cost of treatment of cataract per year of mild/moderate impairment	20 USD	Expert estimates
*CTS*	Cost of treatment of cataract per year of severe impairment	20 USD	Expert estimates
*CTB*	Cost of treatment of cataract per year of blindness	20 USD	Expert estimates
*L_a_*	Life expectancy of a person in age a, a = 40..99	Linear interpolation between values	[18]
*t_m_*	Duration of cataract with mild/moderate impairment	10 years	Expert estimates
*t_s_*	Duration of cataract with severe impairment	5 years	Expert estimates
*t_b_*	Duration of cataract with blindness	Until end of life	Expert estimates
*GDP*	Gross domestic product per capita per annum	1100 USD	[17]
*r*	Interest rate	5%	standard
*CPM*	Productivity loss per year for a person with cataract with mild/moderate impairment	55 USD, i.e., 5% productivity loss	[1,7,8]
*CPS*	Productivity loss per year for a person with cataract with severe impairment	330 USD, i.e., 30% productivity loss	[1,7,8]
*CPB*	Productivity loss per year for a person with cataract with blindness	660 USD, i.e., 60% productivity loss	[1,7,8]
*CCM*	Productivity loss of a caregiver per year for a person with cataract with mild/moderate impairment	0 USD, i.e., 0% productivity loss	[1,7,8]
*CCS*	Productivity loss of a caregiver per year for a person with cataract with severe impairment	55 USD, i.e., 5% productivity loss	[1,7,8]
*CCB*	Productivity loss of a caregiver per year for a person with cataract with blindness	275 USD, i.e., 25% productivity loss	[1,7,8]
*P*	Pension age	60 years	[17]
*s*	Year of surgery	At year of onset, beginning of severe impairment or beginning of blindness	Assumption for scenarios
*CS*	Cost of surgery	300.00 USD	[19,20,21]
*Qm*	Quality of life of a person with cataract with mild/moderate impairment	0.7	[16,22]
*Qs*	Quality of life of a person with cataract with severe impairment	0.6	[16,22]
*Qb*	Quality of life of a person with cataract with blindness	0.5	[16,22]
*Qo*	Quality of life of a person with cataract after operation	0.9	[16,22]

**Table 3 healthcare-10-02580-t003:** Results of baseline scenario (net-values for a = 50) [USD].

Parameter	Without Surgery	With Surgery
Net-value treatment	217.97	162.16
Net-value productivity loss patient	445.93	445.93
Net-value productivity loss caregiver	1,384.54	0.00
Cost operation	0.00	193.38
Total net-value cost:	2048.45	801.47
Cost saving	1246.98	
Quality of life [QALY]	9.59	12.22
Benefit [QALY]	2.63	

**Table 4 healthcare-10-02580-t004:** Thresholds of parameters.

*Variable*	Definition	Original Value	Cost-Saving Threshold	Cost-Effectiveness Threshold	
				≤1∙GDP	≤2∙GDP
*CTM*	Cost of treatment of cataract per year of mild/moderate impairment	20 USD	-	-	-
*CTS*	Cost of treatment of cataract per year of severe impairment	20 USD	-	-	-
*CTB*	Cost of treatment of cataract per year of blindness	20 USD	-	-	-
*L_a_*	Life expectancy	77 yrs.	73	61	
*t_m_*	Duration of cataract with mild/moderate impairment	10	15	-	-
*t_s_*	Duration of cataract with severe impairment	5	-	-	-
*GDP*	Gross domestic product per capita per annum	1100 USD	220 USD p.c.	25 USD p.c.	
*CPM*, *CPS*, *CPB*	Productivity loss of patient	55.00 USD, 330.00 USD, 660.00 USD	-	-	-
*CCM*, *CCS*, *CCB*	Productivity loss of a caregiver	0.00 USD, 55.00 USD, 275.00 USD	-	-	-
*P*	Pension age	60	-	-	-
*CS*	Cost of surgery	300.00 USD	630.00 USD	1800.00 USD	

## Data Availability

The relevant calculations can be obtained from the author.

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
