# Peer review of "Cataract Surgery in Low-Income Countries: A Good Deal!"

_healthcare, 2022, doi:10.3390/healthcare10122580_

Round 1
Reviewer 1 Report
Thank you for you manuscript.
The paper is clearly written, with a robust methodological approach and significant implications for health care policies in the developing world.
One area that may benefit from improvements is the literature support. The evidence quoted for developing countries on the impact of cataract surgery is too thin (just 2 papers) and rather outdated (2010). More recent studies to consider:
Mailu, E. W., Virendrakumar, B., Bechange, S., Jolley, E., & Schmidt, E. (2020). Factors associated with the uptake of cataract surgery and interventions to improve uptake in low-and middle-income countries: A systematic review. PLoS One, 15(7), e0235699.
Ramke, J., Evans, J. R., Habtamu, E., Mwangi, N., Silva, J. C., Swenor, B. K., ... & Ono, K. (2022). Grand Challenges in global eye health: a global prioritisation process using Delphi method. The Lancet Healthy Longevity, 3(1), e31-e41.
McCormick, I., Butcher, R., Evans, J. R., Mactaggart, I. Z., Limburg, H., Jolley, E., ... & Zhang, X. J. (2022). Effective cataract surgical coverage in adults aged 50 years and older: estimates from population-based surveys in 55 countries. The Lancet Global Health.
Author Response
Reviewer 1
- Comment 1
- Reviewer: “Thank you for you manuscript. The paper is clearly written, with a robust methodological approach and significant implications for health care policies in the developing world.”
- Response: Thank you very much for your effort to review this paper and your positive assessment.
- Comment 2
- Reviewer: “One area that may benefit from improvements is the literature support. The evidence quoted for developing countries on the impact of cataract surgery is too thin (just 2 papers) and rather outdated (2010). More recent studies to consider: …”
- Response: We fully agree that the paper requires more up-to-date literature. We actually focused in our literature review mainly on costing studies – and there the literature is limited. But you are very right that we should add some more references conc. up-take and coverage of cataract surgery.
- Action: We added these three references and some more which we had not included before because we assumed that they were outdated. By widening our literature research, we found some papers on cost-effectiveness of surgical interventions in low-income countries which included cataract. However, the costing data of these papers comes from publications which are even olden than what we had cited before. There is a major need to do precise costing!
Reviewer 2 Report
dear author,
The abstract is somewhat vague and needs rewriting, also you must add methods.
discussion: need to add some relevant references that supported your findings.
with regards,
Author Response
Reviewer 2
- Comment 1:
- Reviewer: “dear author”
- Response: Thank you very much for your favourable review and your work in vested into our paper.
- Comment 2
- Reviewer: “The abstract is somewhat vague and needs rewriting, also you must add methods.”
- Response: We fully agree that the abstract has to be improved.
- Action: We re-wrote the abstract and added the methods section.
- Comment 3
- Reviewer: “discussion: need to add some relevant references that supported your findings.”
- Response: We agree that some more references are helpful. Unfortunately, there is nothing new about costing of cataract, but we added more on up-take and coverage.
- Action: We added several references we had not included before because we assumed that they were outdated. By widening our literature research, we found some papers on cost-effectiveness of surgical interventions in low-income countries which included cataract. However, the costing data of these papers comes from publications which are even olden than what we had cited before. There is a major need to do precise costing!
Reviewer 3 Report
Thank you for the opportunity to review this study on cost-effectiveness analysis of cataract surgery in low-income countries.
The idea of this work looks very promising, but the method should be reviewed in depth.
For example, it would be better to use a model (e.g. Markov model) rather than listing different formulas. All parameters used in the model should be listed in a single table with the base value, sensitivity range (for the tornado diagram) and distribution for each parameter. It is necessary that the paper follow the CHEERS checklist (https://www.equator-network.org/reporting-guidelines/cheers/). Some parts need to be added or modified to comply with the methodological expectations of an economic evaluation (examples of problems detected: absence of a probabilistic sensitivity analysis, confusion between quality of life, QALYs, and utilities, no explicit mention of perspective, no presentation of results in a cost-effectiveness plan, no detailed results for each country included in the analysis...).
At this stage, unfortunately, I cannot give a favorable opinion to this draft. I nevertheless encourage you to rework the methods.
Author Response
Reviewer 3
- Comment 1:
- Reviewer: “Thank you for the opportunity to review this study on cost-effectiveness analysis of cataract surgery in low-income countries. The idea of this work looks very promising”
- Response: Thank you very much for the deep throughs which you invested into our paper. We are grateful for this opportunity to add information. However, it seems that we did not express ourselves sufficiently correct. This is not an empirical study as such. We did not go to one particular hospital or intervention program and collect costing data there. In other studies, we did exactly that and there is was much easier to follow what you recommended. However, we are very grateful for the opportunity to explain more about the characteristics of our study.
- Comment 2:
- Reviewer: “but the method should be reviewed in depth. For example, it would be better to use a model (e.g. Markov model) rather than listing different formulas.
- Response: The calculation of a net-value of costs and utilities is a standard methodology in health economic evaluation. A Markov model is a prognostic model which we use if certain developments are stochastic (transition probabilities between stages) and these probabilities are known. If very little is known, it is easier and more transparent to calculate the net-value of a standard patient on a standard pathways. Example: We know the rest life expectancy of a person of a certain age, but we do not have the respective mortality rates for each person in each year. We could, certainly, calculate the mortality rates per year but this would not give us more insights. We decided to assume an average person with a rest life expectancy at on-set of his disease. As we do not look into distributions but averages, this is very much appropriate. The formulae might look difficult, but in reality, they are rather simple as they merely calculate net-values of costs.
- Action: We explained the methodology more and added to the limitations.
- Comment 3:
- Reviewer: All parameters used in the model should be listed in a single table with the base value, sensitivity range (for the tornado diagram) and distribution for each parameter.
- Response: In table 2 we listed all constants. Variables cannot be listed because they are changed within the model. In absence of better data, we cannot give a sensitivity range or distribution of parameters. Consequently, we simply assumed a certain variation of parameters in the Tornado diagram. Please note, for instance, that the cost per cataract surgery are only estimates. There is no reliable literature on this. For other interventions (in particular in high-income countries) we would make a literature analysis and show the minimum, maximum, average and likely also the distribution of these findings. But for cataract in low-income countries these figures simply do not exists.
- Action: we added this to the limitations.
- Comment 4:
- Reviewer: It is necessary that the paper follow the CHEERS checklist (https://www.equator-network.org/reporting-guidelines/cheers/). Some parts need to be added or modified to comply with the methodological expectations of an economic evaluation (examples of problems detected: absence of a probabilistic sensitivity analysis, confusion between quality of life, QALYs, and utilities, no explicit mention of perspective, no presentation of results in a cost-effectiveness plan, no detailed results for each country included in the analysis...).
- Response: The CHEERS-statement is well-known to us. However, it is tailored towards traditional cost-effectiveness studies with a strong and reliable data basis. Some of the points of CHEERS are not applicable to our study, e.g. “Approach to engagement with patients and others affected by the study”.
Furthermore, we do not agree with some of your expectations:
- We do not see that any health economic evaluation requires a probabilistic sensitivity analysis. This makes only sense if the data exists. In our case, we argue that the result is so highly cost-effective even under strongly changed parameters (e.g. on-set age) that we can state in general that it is very likely to be highly cost-effective.
- At the same time, there are scores of health economic evaluations without a cost-effectiveness plan. We are not presenting the outcomes of a particular project or study. Thus, this plan does not apply to our research.
- Furthermore, we do not include several countries. Thus, we cannot present data for each country.
- We also do not understand your statement about “confusion between quality of life, QALYs, and utilities”. We calculate the net-values of quality of life with and without cataract surgery. To our understanding, this is a traditional QALY format as it has been used many times in literature. We would strongly appreciate if the reviewer made clear which sentences he would like to see changed.
We agree that the perspective was not mentioned properly. We added this.
- Action: We made a number of changes to address the issues you raised as far as they seem appropriate for the special situation of evaluating the cost-effectiveness of cataract surgery in low-income countries. In particular, we went through the CHEERS checklist and made some adjustments.
- We wrote a paragraph about uncertainty and how we handled this.
- We explain the feature of this study without a specific field-intervention in more details (e.g. no evaluation plan, no study population)
- We explained the concept of QALYs in more details.
- We described the perspective(s) and time horizon in more details.
Round 2
Reviewer 3 Report
You have improved the paper. Congratulations!